# Isolation and Identification of Natural Colorant Producing Soil-Borne *Aspergillus niger* from Bangladesh and Extraction of the Pigment

**DOI:** 10.3390/foods10061280

**Published:** 2021-06-03

**Authors:** Maria Afroz Toma, K H M Nazmul Hussain Nazir, Md. Muket Mahmud, Pravin Mishra, Md. Kowser Ali, Ajran Kabir, Md. Ahosanul Haque Shahid, Mahbubul Pratik Siddique, Md. Abdul Alim

**Affiliations:** 1Department of Food Technology & Rural Industries, Bangladesh Agricultural University, Mymensingh 2202, Bangladesh; kowserali2011@gmail.com (M.K.A.); maalim07@yahoo.com (M.A.A.); 2Department of Microbiology & Hygiene, Bangladesh Agricultural University, Mymensingh 2202, Bangladesh; nazir@bau.edu.bd (K.H.M.N.H.N.); dvm41187@bau.edu.bd (M.M.M.); mpravin470@gmail.com (P.M.); dvm47012@bau.edu.bd (A.K.); shahid41192@bau.edu.bd (M.A.H.S.); mpsiddique@bau.edu.bd (M.P.S.)

**Keywords:** natural colorants, filamentous fungi, *Aspergillus niger*, PCR, brown pigment

## Abstract

Natural colorants have been used in several ways throughout human history, such as in food, dyes, pharmaceuticals, cosmetics, and many other products. The study aimed to isolate the natural colorant-producing filamentous fungi *Aspergillus niger* from soil and extract pigments for its potential use specially for food production. Fourteen soil samples were collected from Madhupur National Park at Madhupur Upazila in the Mymensingh district, Bangladesh. The *Aspergillus niger* was isolated and identified from the soil samples by following conventional mycological methods (cultural and morphological characteristics), followed by confirmatory identification by a polymerase chain reaction (PCR) of conserved sequences of ITS1 ribosomal DNA using specific oligonucleotide primers. This was followed by genus- and species-specific primers targeting *Aspergillus niger* with an amplicon size of 521 and 310 bp, respectively. For pigment production, a mass culture of *Aspergillus niger* was conducted in Sabouraud dextrose broth in shaking conditions for seven days. The biomass was subjected to extraction of the pigments following an ethanol-based extraction method and concentrated using a rotary evaporator. *Aspergillus niger* could be isolated from three samples. The yield of extracted brown pigment from *Aspergillus niger* was 0.75% (*w*/*v*). Spectroscopic analysis of the pigments was carried out using a UV–VIS spectrophotometer. An *in vivo* experiment was conducted with mice to assess the toxicity of the pigments. From the colorimetric and sensory evaluations, pigment-supplemented products (cookies and lemon juice) were found to be more acceptable than the control products. This could be the first attempt to use *Aspergillus niger* extracted pigment from soil samples in food products in Bangladesh, but for successful food production, the food colorants must be approved by a responsible authority, e.g., the FDA or the BSTI. Moreover, fungal pigments could be used in the emerging fields of the food and textile industries in Bangladesh.

## 1. Introduction

The addition of food colorants is commonly used to maintain and enhance the real color of a food substance, and sometimes to preserve it as well [1]. Plants are a good source of pigments, but due to some issues such as seasonal dependency, price, and variations in color intensity and hues [2], researchers and industry prefer to use microbes for pigment production. This is due to the ease of large-scale production for the extraction of pigments [3], which are more cost effective and can be harvested throughout the year with no side effects, as well as being eco-friendly and biodegradable [4]. Colorants obtained from microorganisms (yeast, fungi, bacteria, algae, etc.) can be labeled as natural colors due to their origin [5]. Among several microorganisms present in soil, fungi are metabolically active and able to produce many secondary metabolites including pigments which contains different beneficial bioactive compounds with antibiotic, anticancer, antimicrobial, and antioxidant effects [6].

Dyes obtained from nature are eco-friendly and more easily degradable than other synthetic dyes [7]. Because of these beneficial effects, the pharmaceutical and agrochemical industries have an interest in these secondary metabolites which include cephalosporin [8], cyclosporine [9], statin drugs [10] and, most importantly, penicillin [11]. Lovastatins or monacolins produced by fungi such as *Penicillium*, *Monascus*, and *Aspergillus* [12] suppress cholesterol biosynthesis [13]. *Aspergillus* spp. are known to produce anthraquinone which has commercial importance due to its antibacterial and antifungal properties [14]. The most active extracts obtained from *Aspergillus* spp. are subjected to both herbicidal and phytotoxic activities [15]. Synthetic colorants have become more popular for many reasons such as color fastness, a wide range of color availability, low production costs, etc., but these dyes act as a potential source of various diseases for everything from simple skin allergies to cancer [16]. As a means to promote the use of healthier food additives, filamentous fungi are being used under supervision in economically advanced countries as a readily available source of natural colorants [17].

Microbial colorant production is becoming an exciting and important field of interest for research. In nature, such pigment-producing microorganisms are available in a wide range of colors [18]. Terrestrial systems are a rich source of filamentous fungi [19]. Keeping that in mind, the study was developed to conduct research into colorant-producing filamentous fungi in selected areas of Madhupur National Park (MNP), Mymensingh. MNP is located at 24°45′ N latitude and 90°05′ E longitude and encompasses an area of 8436 ha (Figure 1) [20]. The soil color of this forest is mostly reddish, but different categories also exist such as brownish-red, yellow-red, and pale brownish-red due to iron ore containing manganese (Madhupur Kankar) [21]. The pH values of these forest soil samples were ranked from 4.23 to 5.65, which is also responsible for the reddish or yellowish color of the soil. The textural composition of the forest soil was clay and clay–loam [22]. The study was conducted to isolate a fungal variety from Bangladesh which is capable of producing a pigment in a cost-effective way. The best performing strain found was *Aspergillus niger*, which we want to make available for future use in the food industry. The present study could be considered the first conducted on the isolation of *A. niger* from soil and the extraction of natural colorants in the context of Bangladesh.

## 2. Materials and Methods

### 2.1. Collection of Soil Samples and Site Selection

A total of fourteen soil samples were collected aseptically using sterilized enclosed zipper bags with proper labeling from different sites in the Madhupur National Park, Madhupur, Mymensingh, Bangladesh (Figure 1). The collected samples were carried to the laboratory on the same day and kept under normal storage conditions for microbiological studies. The laboratory experiments were performed at the Department of Microbiology and Hygiene, Bangladesh Agricultural University (BAU), Mymensingh, Bangladesh.

### 2.2. Isolation of Fungal Strains from Soil Samples

Solid media of potato dextrose agar (PDA) were used for the isolation and identification of the fungal strains. Inoculum prepared from the soil samples was spread onto PDA media, and incubation was carried out at 28 ± 2 °C for 5–7 days. After incubation, the colony morphology was recorded to identify the specific fungus collected in each sample. Colonies of different fungi were subcultured onto PDA. A small piece of fungal mycelia was picked up by sterile toothpick and directly placed on PDA and then incubated at 28 ± 2 °C for 5–7 days to obtain pure cultures.

### 2.3. Identification of Aspergillus niger

The suspected *A. niger* was identified by staining and molecular detection methods. For staining, the isolate was stained with lactophenol cotton blue (LPCB) to observe the morphological characteristics, such as hyphae, conidial heads, and arrangements under a microscope [23]. For the molecular detection by polymerase chain reaction (PCR), the fungal DNA extraction, PCR primer design, PCR reaction mixture, and thermal profile were maintained as per the methods of Sugita et al. [24]. The DNA was extracted using the phenol–chloroform extraction method, which was used as the template for the PCR amplification of the conserved sequences of the internal transcribed spacer 1 (ITS1) ribosomal DNA and its flanking regions. This was done by using *Aspergillus* spp. genus-specific primers (ASAP-1: 5′ CAGCGAGTACATCACCTTGG 3′ and ASAP-2: 5′ CCATTGTTGAAAGTTTTAACTGATT 3′) with an amplicon size of 521 bp, followed by *A. niger* species-specific primers (ASPU: 5′ACTACCGATTGAATGGCTCG 3′ and Nilr: 5′ ACGCTTTCAGACAGTGTTCG 3′) with an amplicon size of 310 bp. For the negative control of genus-specific primers and species-specific primers, the DNA templates of *Fusarium* spp. and *Aspergillus flavus* were used, respectively.

### 2.4. Pigment Production and Extraction from Aspergillus niger

#### 2.4.1. Primary Culture of *Aspergillus niger*

After cultivation for 5–7 days of the isolated fungus, a small amount of inoculum was collected and mixed with 6–8 mL sterilized water. Later, a small amount of spore suspension was inoculated in 50 mL of submerged culture medium in conical flasks and then cultivated in a rotary shaker incubator at 28 ± 2 °C at 150 rpm for 5–7 days.

#### 2.4.2. Submerged Fermentation of *Aspergillus niger*

Fermentation in liquid media was prepared using Sabouraud dextrose agar (SDA) with distilled water in an Erlenmeyer flask. The initial pH of the liquid media was adjusted to 6 by the addition of HCl or NaOH. Spores from the surface area of the *A. niger* strain culture were inoculated into the fermentation medium. Incubation was done for 5–7 days at 28 ± 2 °C using an agitator for the fermenter at 150 rpm. To optimize the fermentation conditions, SDA powder (30 g), yeast extract (4 g), sucrose (10 g), NaNO_3_ (3 g), K_2_HPO_4_ (1 g), MgSO_4_.7H_2_O (5 mg), KCl (5 mg), and FeSO_4_.7H_2_O (1 mg) were used in 1000 mL (w/v) solution [25].

#### 2.4.3. Pigment Extract Produced by *Aspergillus niger*

Pigment was extracted from the broth by the solvent extraction method [26]. Sabouraud dextrose broth was prepared for the *Aspergillus* culture, and then the inoculate was incubated for 7 days. After incubating the culture, filtration was done to separate biomass from the broth and a micro filter (0.45 µm sterile micro filter, Sartorius stedium biotech, France) was used to filter the supernatant broth, removing the unnecessary fungal substances. A volume of 95% (*v*/*v*) ethanol was added to the filtrate and kept on a rotary shaker incubator for 30 min at 150 rpm and 30 °C [25]. After this, centrifugation was conducted at 5000 rpm for 15 min. The same process was repeated for removing the fungal biomass, and the filtration process was done with Whatman filter paper. The extracted colorant was then concentrated using a rotary evaporator Model No: 2,600,000 (witeg, Labortechnik GmbH, Wertheim, Germany) at 45 °C and 30 rpm to obtain the brown pigment in a semisolid form [27,28].

### 2.5. Detection of Fungal Pigment by Spectroscopic Analysis

The fungal pigment obtained after extraction and drying (1 mg pigment) was dissolved in 4 mL of distilled water to obtain the UV absorption spectrum [29]. The UV–VIS spectrum of the brown pigment sample was obtained using the WTW photoLab^®^ 7600 UV-VIS spectrophotometer (Xylem Analytics LLC, Weilheim, Germany) in the spectral range of 200–700 nm [30].

### 2.6. In Vivo Test for Toxic Analysis of the Produced Pigments

To determine the toxicity of the extracted color, a total of 25 Swiss albino mice (ages ranging from 6 to 8 weeks; body weight ranging from 90 to 120 g) were used following the methods described by Ali et al. [31] and Elekima and Nwachuku [32], with slight modifications. The mice were collected from the Department of Microbiology and Hygiene, Bangladesh Agricultural University, Mymensingh, and were originally purchased from the International Centre for Diarrhoeal Disease Research, Bangladesh (ICDDR’B). The mice were divided into 5 groups comprising 5 mice each. The groups were housed separately and adapted there for a week. After adaptation, different doses of extracted color, such as 20 µL, 40 µL, 60 µL, 80 µL and a negative control (only water), were applied to the 5 groups of mice using a micropipette [23]. All the mice were supplied with water and a standard diet (pellet; rat premix) throughout the toxicity test period (28 days). Under regular supervision, the mice were checked out for the 28 days to see whether they had developed any unexpected symptoms or died (Figure 2). An ethical statement was taken from the animal welfare authority of Bangladesh Agricultural University, Mymensingh, to ensure that no animals were harmed during this study. The ethical statement number is AWEEC/BAU/2019(33).

### 2.7. Supplementation of Fungal Colorants in Food Products

#### 2.7.1. Cookies

For preparing the cookies, 100 g wheat flour, baking powder, fat, sugar, and egg whites were added into a prepared creamed mixture along with 1% of the obtained brown color and 2–3 drops of vanilla essence. A small amount of milk was added to prepare the dough kept in a tray to make a roll of the desired thickness. The roll was then cut into small pieces and baked for 20 min at 180 °C [25].

#### 2.7.2. Lemon Juice

For preparing the lemon juice, fresh and ripe lemons were used. After washing, each lemon was sliced and transferred into a hand juicer to extract the juice. Extracted juice was filtered using muslin cloth and mixed with 0.25% of the obtained brown color then blanched at 80 °C for 5 min with the required amount of sugar and cooled immediately [33].

### 2.8. Assessment of Supplemented Fungal Colorant on Food Product Quality

#### 2.8.1. Quantitative Colorimetric Analysis

To individualize any color using the CIE L* a* b* color system, a chroma meter (CR 400, Konica Minolta INC. Japan) was used. L* defines the lightness (dark to light color ranges from 0–100), a* value defines the attributes of color (blue to yellow ranges from −60 to +60), b* value defines the vividness or dullness, C* describes the chroma expressed as red or green values of color, and h denotes the hue angle of the color. Hue angle values also indicate the degree of redness, yellowness, blueness, and greenness [34].

#### 2.8.2. Sensory Evaluation

A taste panel of ten semi trained panellists evaluated the cookies and lemon juice according to the score card method to assess sensory parameters such as color, texture, flavor, taste, and overall acceptability. A 9-point hedonic rating test was used for the statistical analysis to assess the degree of acceptability of the products [35].

### 2.9. Statistical Analysis

The analysis for the study was performed three times. The software STATISTIC version 8.1 was used to assess the significance of the different mean values. DMRT was also applied to assess the significance of these differences.

## 3. Results

### 3.1. Results of Isolation and Identification of Aspergillus niger

#### 3.1.1. Isolation of Fungal Strains and Cultural Characterization of *Aspergillus niger*

Potato dextrose agar (PDA) was inoculated with the inoculum prepared from 14 soil samples, and 10 samples revealed a positive based on the colonial growth of fungi after 5–7 days of incubation at 28 ± 2 °C. After a subculture of the obtained filamentous fungi was taken, several different pigment-producing fungi were found. Among those heterogeneous fungal isolates, *A. niger* was chosen from which to extract a natural colorant. The colony of *A. niger* was initially white and then turned into a variety of shades of yellow, green, brown, or black, with a velvety or cottony texture (Figure 3).

#### 3.1.2. Identification of *Aspergillus niger* by Observing Microscopic Morphology

The colony was observed by the morphological characteristics of *A. niger*. After 4 days of incubation period at 28 °C, *A. niger* produced colony diameters of 40–50 mm. The initial growth was white and later became black, which was the result of darkly pigmented conidia. The reverse of the colony was pale yellow in color. The conidial heads of *A. niger* were found to be large and dark brown as observed under a microscope. Conidiophores were dark in color towards the globule. The conidial heads were in a form of biseriate brown but often with separate metulae. The vesicles were globose, dark brown, and rough walled (Figure 4).

#### 3.1.3. Identification of *Aspergillus niger* by Polymerase Chain Reaction

DNA extracted from culture-positive isolates was subjected to polymerase chain reaction (PCR) assays. PCR with ASAP-1 and ASAP-2 primers confirmed the isolates were *Aspergillus* spp. (*n* = 4; 28.57%), showing an amplification of 521 bp. The isolate as *Aspergillus* sp. was confirmed by using PCR with primers ASPU and Nilr, which confirmed the isolates were *A. niger* (*n* = 3; 75%) as amplifications were seen at 310 bp (Figure 5).

### 3.2. Results of Aspergillus niger Fermentation

Nine days was the optimum time for the fermentation of *A. niger* needed to obtain a natural pigment from a liquid solution with a pH of 4.5. The yield of natural colorant from *A. niger* was 0.75% (*w*/*v*) (Figure 6). Continuous agitation by rotary shaker at 150 rpm was maintained for a better fermentation process. To optimize the fermentation conditions of *A. niger*, Sabouraud dextrose broth was used with different sources such as carbon, nitrogen, and salts. From the mycelia of *A. niger*, the brown-colored pigment was isolated at the end of the incubation period.

### 3.3. Spectroscopic Analysis of Brown Pigment and Absorbance in Solution

The UV-visible spectrum of purified pigment showed a broad absorption spectrum from UV to the visible region, and the absorption was higher in the UV compared to visible region (Figure 7). Four different concentrations (0, 0.5 mL, 1.0 mL, and 2 mL) of color were dissolved in 50 mL of distilled water. After that, the absorbance of the solutions was measured by spectrophotometer at 295 nm wavelength. The absorbance reading at different concentrations is shown in Figure 8.

### 3.4. Results of Toxic Analysis Test

In the study, only *in vivo* (mice) tests were used for toxic analysis. A total of 5 different doses were applied to the mice for 28 days. The mice were alive for 28 days and no abnormal activities were observed during this period.

### 3.5. Quantitative Colorimetric Analysis of Pigments in Food Products

Figure 9 shows the L*, a*,b*, C*, and h of the cookies and lemon juice before and after pigment dilution. The L* values decreased with the addition of the pigment into the cookies and lemon juice. The cookies without pigment showed a higher L* value (70.75) than those with pigment (56.91). The addition of the pigment into the lemon juice decreased the L* values compared to the control value. Both the a* and b* values increased with the addition of the pigment into the cookies. The increase in a* and b* stimuli was due to the addition of the colorant which increased the redness and yellowness. In the lemon juice, the a* and b* values also increased with the addition of the pigment (Figure 10).

### 3.6. Sensory Evaluation of Fungal Pigment-Supplemented Food Products

ANOVA performed on the sensory attributes of the cookies and lemon juice with or without colorants showed that there was a significant (*p* < 0.5) difference in the sensory attributes (color, flavor, texture, taste, and overall acceptability) between the control and pigmented groups (Table 1). Colored food products were judged to have a better color, flavor, texture, taste, and overall acceptability than the control samples. The overall acceptability scores of the colored cookies (8.48 ± 0.35) and the colored lemon juice (8.08 ± 0.30) showed that the products were more acceptable than the colorless samples of cookies (7.92 ± 0.19) and lemon juice (7.55 ± 0.23). These values were expressed as mean ± SD (Table 1). It is fervently hoped that in the future, this fungal pigment might receive greater attention from the food industry.

## 4. Discussion

The present aptitude of the community for using natural products has been increased due to the toxic and carcinogenic effects of synthetic ones [36]. Novel techniques for the production of food colorants are being discovered and increasing day by day [37]. Soil samples were cultured using the potato dextrose agarose (PDA) and the fungi were primarily isolated based on colonies after 7 days of incubation which was also reported by Ravimannan et al. [38]. After subcultures of the obtained filamentous fungus *A. niger* were found and chosen, research continued and produced findings similar to those of Devi [39]. *A. niger* produced colony diameters of 40–50 mm and the initial growth was white. The reverse of the colony was a pale yellow color based on the findings of Ray and Yakin [40]. Ponraj et al. also reported aseptate, short conidiophores, and terminal globose vesicles, as well as doubled sterigmata covered with vesicles and cottony growths of green or yellow with black spores [41]. DNA extracted from *A. niger* culture-positive isolates were used in the first PCR assay. Molecular detection of *Aspergillus* spp. was carried out by PCR using genus-specific primers ASAP-1 and ASAP-2 which amplified 521-bp fragments. Such an effect was also similarly described by Sugita et al. [24]. For optimizing the fermentation conditions of *A. niger*, Sabouraud dextrose broth (SDB) medium was used with different sources such as carbon, nitrogen, and salts. Guyomarch et al. reported the use of different nutrient sources in culture media to optimize the fermentation conditions [42].

From mycelia of *Aspergillus niger*, the brown-colored pigment was extracted from the submerged culture. Atalla et al. [43] found the same pigment produced at the end of the incubation period of *A. niger*. Flavasperone is a pale-yellow pigment which was isolated from the mycelium of *A. niger* for the first time in this study. There were also some other colors of *A. niger* which were observed as black by Ray and Yakin [40]. The pigment was then characterized by a UV–VIS spectrophotometer. In the spectrum, the pigment gave two peaks: one at around 250 nm and another at 295 nm wavelength. Generally, the brown pigment has greater absorption in the UV region which decreases progressively with an increase in wavelength (300–700 nm), showing a linear correlation between absorption and wavelength. This is a characteristic feature of brown pigments which was similarly described by Goncalves et al. [29]. The pigment was both soluble in water and alcohol. The absorbance was measured by spectrophotometer.

The extracted pigment went through a toxic assessment test by *in vivo* (mice). A total of 5 different doses were applied to the mice for 28 days, as per a previous report by Bechtold [44]. The mice were found alive, and no abnormal activities were observed during that period. However, before consuming or declaring the colorant as food grade and safe for human consumption, the colorant should be analyzed by more advanced technologies to ensure whether the pigment is safe or not.

After this test, a colorimetric analysis of the pigment-supplemented food products was conducted by a colorimeter. For the lemon juice, a* and b* values were found to have increased with the addition of the pigment. The C* value increased and h value decreased with the addition of colorant. This means that the redness increased in both the values of the lemon juice and the cookies, which was the same as the findings of Zalar et al. [45] and Sutthiwong et al. [46]. The sensory evaluation of the pigment-supplemented food products was conducted by observing the organoleptic properties (color/appearance, flavor, texture, and taste) by panelists. The pigment-supplemented products were significantly (*p* < 0.5) more acceptable by the panelists than the control products (without colorant).

## 5. Conclusions

Today, people are more health conscious and have knowledge about natural products. This is why natural colorant production and application have increased in the past few years. *Aspergillus niger* was isolated from soil and a brown-colored pigment was extracted from the submerged cultivation of the isolate. In this research, 0.75% (*w*/*v*) semiliquid color was obtained per liter of fermented solution of *A. niger*, which is an appreciable amount, and the pigment-supplemented food products were found to be more acceptable by the panelists than the control food products. More research on fungal pigment from *A. niger* needs to be conducted to elucidate the chemical structure with up-to-date analytical techniques such as HPLC, FTIR, and NMR. Moreover, the biosafety of the pigment must be investigated to allow for its use in food applications.

## Figures and Tables

**Figure 1 foods-10-01280-f001:**
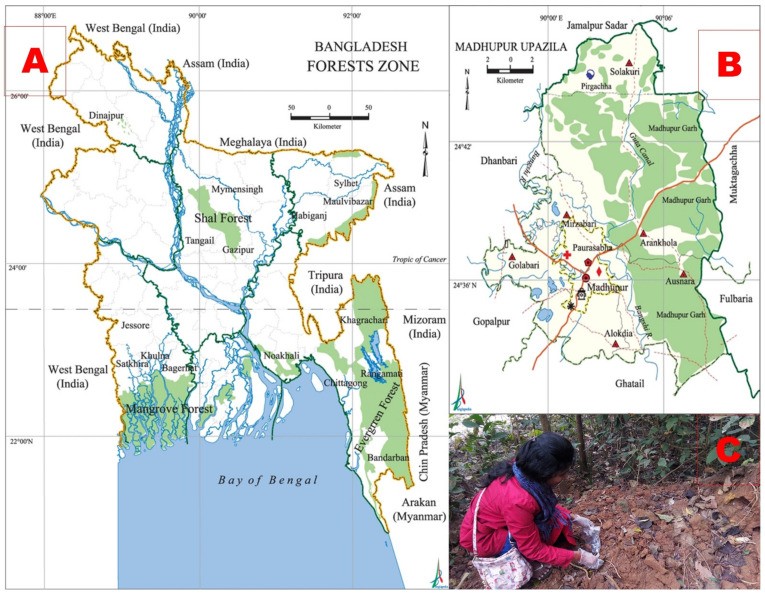
Pictorial view of sampling area ((**A**): map showing Bangladesh; (**B**): map showing Madhupur National Park; (**C**): sampling). Courtesy of http://en.banglapedia.org/index.php/Forest_and_Forestry (Accessed on 1 May 2021).

**Figure 2 foods-10-01280-f002:**
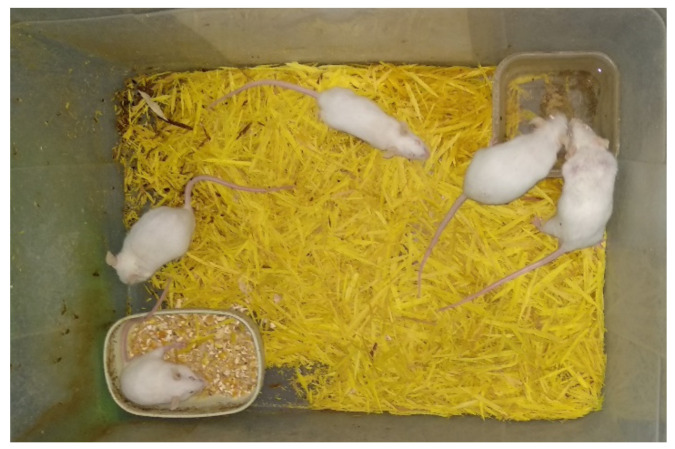
A group of Swiss albino mice after 28 days of color feeding.

**Figure 3 foods-10-01280-f003:**
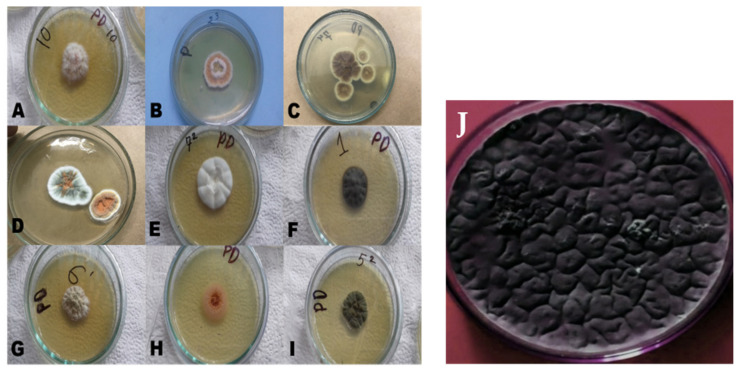
Different unidentified soil-borne fungi (**A**–**I**) and a cultural characterization of *Aspergillus niger* (**J**).

**Figure 4 foods-10-01280-f004:**
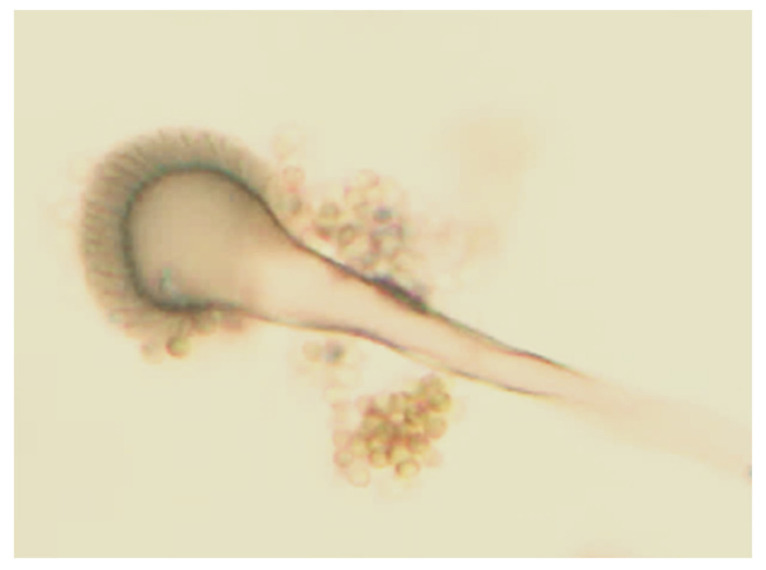
Microscopic morphology of *Aspergillus niger* viewed at 100×. The morphology of *Aspergillus niger* shows large, globose, dark brown conidial heads. Conidiophores were smooth walled, hyaline, or turning dark towards the vesicle.

**Figure 5 foods-10-01280-f005:**
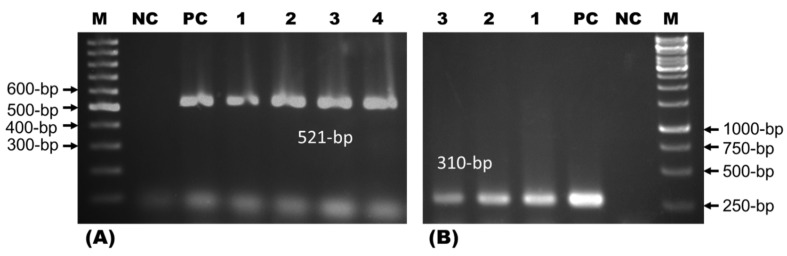
Identification of *Aspergillus* sp. and *Aspergillus niger* by polymerase chain reaction. (**A**) PCR identification of *Aspergillus* sp. by using genus-specific primers. Lane M- 100-bp DNA ladder, NC: negative control, PC: positive control, and lane 1–4 test samples. (**B**) PCR identification of *Aspergillus niger* by using specific primers. Lane M—1 Kb DNA ladder, NC: negative control, PC: positive control, and lane 1–3 test samples.

**Figure 6 foods-10-01280-f006:**
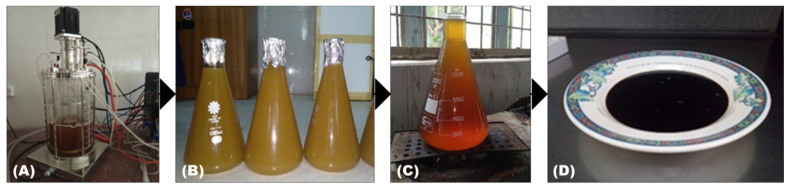
Pigment extraction produced by *Aspergillus niger*. (**A**) Fermentation by *Aspergillus niger*; (**B**) filtrate after fermentation; (**C**) heat treatment of the filtrate; (**D**) final product found after rotary evaporation.

**Figure 7 foods-10-01280-f007:**
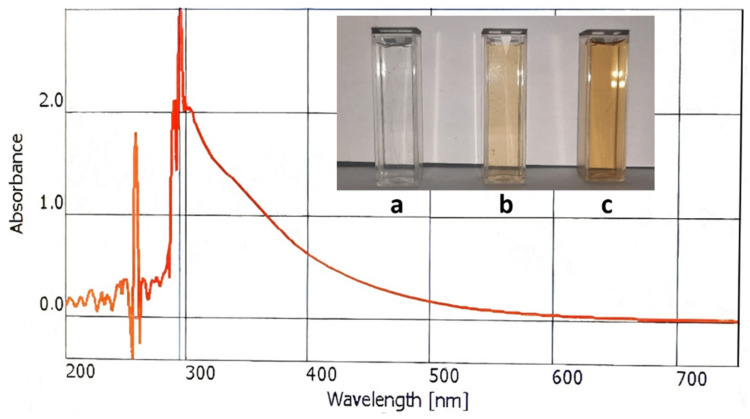
UV-visible spectra of extracted fungal pigment; highest absorbance observed at 295 nm. (**a**) Negative control: distilled water; (**b**) 0.50 mL pigment diluted with 50 mL distilled water; (**c**) 1.0 mL pigment diluted with 50 mL distilled water.

**Figure 8 foods-10-01280-f008:**
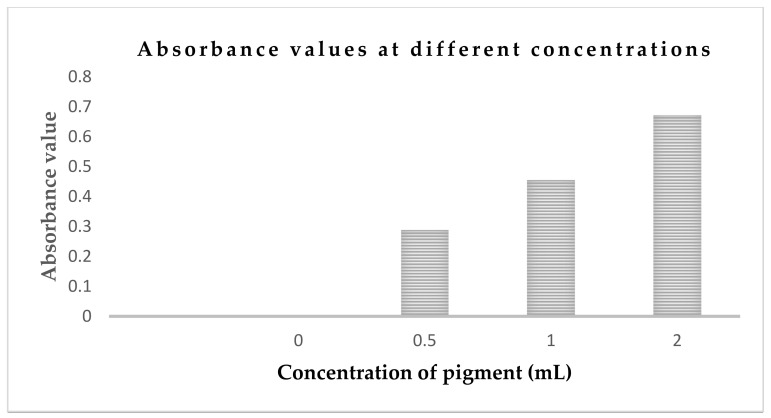
Absorbance values at different concentrations (0, 0.5 mL, 1.0 mL, and 2 mL) of diluted pigments in 50 mL of distilled water.

**Figure 9 foods-10-01280-f009:**
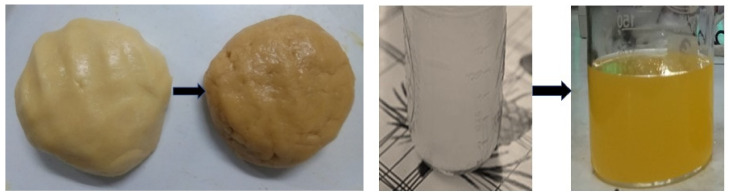
Addition of extracted pigment with cookie dough and lemon juice.

**Figure 10 foods-10-01280-f010:**
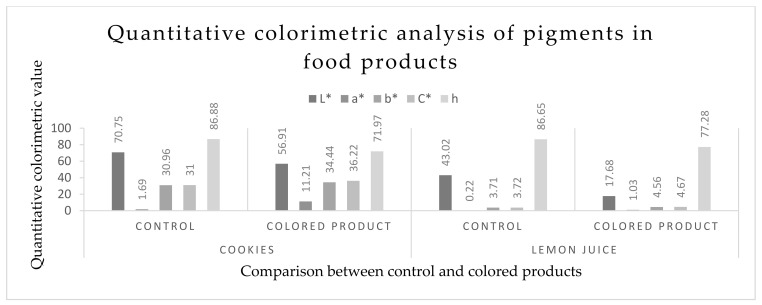
Comparison of L*, a*, b*, C*, and h between colored and colorless cookies and lemon juice, where, L*—lightness, read from 0 (black) to 100 (white); a*—(positive) red color, (negative) green color; b*—(positive) yellow color, (negative) blue color; C*—purity of the color; h—hue.

**Table 1 foods-10-01280-t001:** Sensory evaluation of the fungal colorant-supplemented food products.

Sensory Parameters	Sensory Scores				LSD at 5%	Standard Error of the Mean (S.Em±)
	Cookies		Lemon Juice			
	Control	with Colorant	Control	with Colorant		
Color and appearance	6.89 ± 0.21	8.54 ± 0.53	6.51 ± 0.71	8.22 ± 0.38	0.534	0.601
Flavor	8.23 ± 0.43	8.36 ± 0.38	8.20 ± 0.75	8.38 ± 0.50	0.621	0.689
Texture	8.47 ± 0.10	8.61 ± 0.36	7.50 ± 0.45	7.57 ± 0.43	0.378	0.554
Taste	8.07 ± 0.25	8.40 ± 0.53	7.99 ± 0.91	8.13 ± 0.53	0.450	0.475
Overall acceptability	7.92 ± 0.19	8.48 ± 0.35	7.55 ± 0.23	8.08 ± 0.30	0.499	0.583

Values are expressed as mean ± SD. Least significant difference at 5% (*p* ≤ 0.05) based on ANOVA.

## Data Availability

Data available on request.

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
