# Peer review of "Isolation and Identification of Natural Colorant Producing Soil-Borne Aspergillus niger from Bangladesh and Extraction of the Pigment"

_foods, 2021, doi:10.3390/foods10061280_

Round 1
Reviewer 1 Report
Dear authors,
I liked your manuscript, because it is innovative and uses a new strain for the production of a natural dye as colorant. However, I would like to see several improvements of the manuscript.
In a manuscript which is dealing about coloring of food products it would be mandatory for me to show pictures of the colored products compared to the no colored products. I found some discrepancies in the text were you sometimes write that you have a strain collection and sometimes only write about the Aspergillus strain. Moreover, you sometimes did not write the species name in italic!
You mixing up British and American English e.g. you sometimes write color and sometimes colour please make it consistent.
I try to summarize all my critics one by one:
In lines 2-3: I would recommend changing the title to: Isolation and Identification of Natural Colorant Producing, Soil-born Aspergillus niger from Bangladesh and Extraction of the Pigment
In line 15: the N in Natural should not be bold
In lines 28-29: “Detection of the pigments was done by using UV-VIS Spectrophotometer.” You can’t simply identify a pigment by using UV-VIS. Do you mean you identify the absorption maximum? Moreover, spectrophotometer should be written with a small s
In line 34: Aspergillus niger should not be not abbreviated
In lines 37-38: you should add that food colorants have to be approved by responsible authority e.g. FDA or EFSA
In lines 38-39: I would recommend so write: “Plants are a good source of pigments, but due to some problems, such as seasonality, high price and variations in color intensity and hues…”
In line 44: I would suggest to write natural colors instead of bio colors
In lines 46-47: I would suggest to write “bioactive compounds with antibiotic, anticancer, antimicrobial and antioxidant effects”; after pigments and before which add a comma please
In lines 49-50: I would suggest to write “Because of these beneficial effects, the pharmaceutical and agrochemical”; moreover I’m not sure what do you mean with stain drugs? Do you mean statins?
In line 52: please change emanated to derived or produced
In line 53 and line 55: it should be “Aspergillus spp. are known …..”
In lines 56-58: it should be “color fastness” and please add commas after cost and etc.,
In lines 65-66: the geographical coordinates should have the same format; latitude and longitude both not with capital letters; moreover I would suggest to write “encompass a notified area of 8,436 ha”; no point after ha
In line 69: it should be “iron ore containing manganese”
In line 70: I recommend not starting the sentence with pH, maybe “The pH-values of these forests soil rank from 4.23 to 5.65, which …”
In lines 72-73: I would suggest rephrasing the sentence “The study was conducted to isolate A. niger from local isolate of Bangladesh so that the research would be cost effective and available for future prospective in food industry.” To “The study was conducted to isolate a fungal isolate from Bangladesh, that is capable to produce a pigment in a cost effective way. The best performing strain was Aspergillus niger, which we wanted to make available for future prospective in food industry.”; in the main text Aspergillus niger was not mentioned before so it should not be abbreviated here
In line 78: soil with small letters
In lines 85-86 and figure 1: I would recommend to “split” in the Figure 1 in parts (A, B and C) and write a more informative figure caption.
In line 87: I suggest to write only “Isolation of fungal strains from soil samples”
In line 88: you write “Solid media such as Potato Dextrose Agar (PDA)” did you use other media, which were not mentioned, than mention please otherwise I would recommend to change the sentence e.g. be remove “such as”
In line 89: please write fungal strains or fungal species
In line 92: please change “A small lump of …” to “A small piece of …”
In lines 95-96: Aspergillus niger has to be not italic, because the rest part of these (sub)heading is italic, while in text A. niger has to be italic
In line 102: add a comma after method and after was there are two spaces
In line 110: change to “Pigment production and extraction from A. niger”
In line 112: it should be fungus not fungi as fungi is the plural form; please use mL
In line 118: “The initial pH of the liquid medium was adjusted to 6 …”
In line 119: “A. niger strain was inoculated into the fermentation medium.” This is not clear for me how did you add the fungus? As spores or as mycelium, how much did you add?
In lines 120-121 please use a protected space between 150 and rpm
In lines 121-123: yeast extract with small letters; add spaces between the digits and the units, please use mL;
In lines 127-128: I recommend to write “After incubating the culture, filtration was done to separate biomass from the broth a micro filter used.”; please add more information about the micro filter (supplier and place, filter size e.g. 0.4 µm micro filter)
In line 130: space between 30 and °C
In line 131: please give the rotation speed in multiple earth force instead of rpm
In line 133: please change: “using rotary evaporator Model No: 2 600 000 (witeg Labortechnik GmbH, Wertheim, Germany) “ and 45 °C
In line 134: I did not 100% understand why you cite these references here? As I understand, they did not use an Aspergillus strain in the cited work; moreover, it is clear that after the evaporation you will get most likely solid
In line 135: small letters for spectroscopic
In line 136: Currently I didn’t get if you have purified the pigment or not, I guess you only extracted it and removed the solvents, that’s why I would recommend to write “The obtained fungal pigment after extraction and drying ..”; please use 4 mL
In line 138: using the WTW photoLab® 7600 UV-VIS spectrophotometer (Xylem Analytics LLC, Weilheim, Germany)
In line 141: please give more information how the mice were breeded! I recommend to rephrase the sentences here like: “The biological toxicity of the produced pigment was evaluated in mice (according to the appropriate and recognized ethics) as follows: age + gender mice (white mice, race e.g. Mus musculus domesticus) weighing x g were obtained from xxxx”; The animals were provided free access to water and food?
In line 142 and 151: a space between 40 and µL or 100 and g
In line 155: Lemon juice
In line 161: Quantitative colorimetric analysis
In line 171: do you mean visual color by viz. color? Please change that
In line 177: assess instead of asses
In line 179: I would recommend that you change the heading here, because you should write first about the isolation of fungal strains from the soil samples before you go to your candidate A. niger
In line 182 it should be: “fungi were primarily isolated”
In lines 183-184: “After sub-culture of the obtained pure filamentous fungus strains, different soil born strains were identified producing pigments.” I would recommend to add a picture of at least the fungi, which produces pigments on the PDA plates here. Moreover, I would recommend to add some numbers here, like how much fungi did you isolated in total? Did you get from each of the fourteen soil samples the same amount of fungal strains?
In line 185: “The colony of Aspergillus niger was initially white and then any shade 185 of yellow, green, brown, or black with velvety or cottony texture.” please check that it is not italic.
In line 191: use a protected space between A. and niger
In lines 202-203: “Aspergillus sp. (n=4; 28.57%) showing amplification of 521 bp. That our isolate belonging to Aspergillus sp. was confirmed by subjected PCR with primer ASPU & Nilr, which ….”
In lines 206ff: in the figure caption I would recommend you to make it consistent e.g.: NC- negative control, PC: positive control; lane should be written with small l; what was the negative control it wasn’t mentioned in the M&M part as well
In the line 210-218: the heading is “Results of A. niger fermentation”, while at the end you write “After 9 days of fermentation, the four different pigments i.e., yellow, green, brown, and orange were obtained from four different fungi.” please make it consistent.
I would appreciate if you would write the IDs you gave the strains as well.
In lines 220-221: please change the caption of the figure, for me it is not 100% clear what you show here
In line 222: “The UV-VIS spectrum of extracted pigment …”
In line 224: 0 µL
In line 227: “The absorbance reading at different concentration is shown in Figure 6.” This sentences don’t make sense please change it
In line 229: UV-VIS spectra; explain in the captions what is a, b and c.
In lines 230-231: I would recommend to change the figure maybe to bar plots with these circle plots you normally illustrate something with is summable; the figure captions has to be updated
In lines 232ff: I would recommend to change it “In the study, only in vivo (mice) tests were done to analyzing toxic effects.” Please add point after period. Moreover, it would be good to show pictures of the mice and if you have add more data here e.g. body weight. I could not really find out how the pigment was applied to the mice. With subcutaneous injection? Do you have pictures of the mice? As recommendation for upcoming experiments you show think about to cooperate with somebody how makes more experiments in this direction. It would be good to see the effects on liver, kidney, heart and spleen.
In lines 251ff: I would suggest do better describe the effects on color, flavor, texture, taste and overall acceptability. For my point of view the flavor, texture, taste are similar to the control, while color and overall acceptability is much higher. I would recommend to add pictures of the colored products and controls here was well! Just for my personal interest: Why did you wanted to color lemon juice black?
In line 261-262: “The present aptitude of the community for getting natural product has been increased.2 I would recommend to change that sentences a) to sound more scientific and b) add a reference here and c) give reasons for that.
In line 293ff: with your experiments you can not role out that the pigment “can assume to be safe for human consumption”, of course you write that more experiments have to be done, but as you didn’t perform a proper biosafety evaluation of the mice e.g. by inspection of changes of the organs you can only state that no abnormal activities observed or death were observed!!
In lines 308-309: I recommend to change the sentence to “For this reason, the production and use of natural dyes has increased in recent years.”
In line 315: “Due to lack of funding and equipment, the study didn’t go so far.” Remove this sentence I can image that you don’t have enough money to perform more experiments but this sentences should not be in scientific manuscript!
In lines 316-317: The sentences “The authors want to do more research on fungal pigment in future with more advance 316 technology such as HPLC, FTIR, NMR etc. to purify the pigment.2 did not sound scientifically correct. Please change it such as “More research on fungal pigment from A. niger has to be done do elucidate the chemical structure with up to date analytical techniques such as HPLC, FTIR and NMR. Moreover, the biosafety of the pigment has to investigate to allow its use for food applications.”
Author Response
Dear Reviewer,
Thank you so much for giving us your precious time. We have edited our manuscript as per your recommendation. Hope that you won't be disappointed.
Kind regards.

Reviewer 2 Report
The manuscript is well written and easily understandable. All the data are represented clearly and precisely. Authors should read this manuscript carefully and correct minor errors such as scientific nomenclature.

Author Response
Dear Reviewer,
Thank you so much for giving your precious time to the manuscript and also for appreciating our works. We have checked the manuscript thoroughly and edited as per your given comments.
Kind regards.
Round 2
Reviewer 1 Report
Dear authors,
Thank you for the many changes. Now the manuscript is in a much better state.
I still have two small points. in line 161 it should be g and not gm.
The second point is that maybe my mistake is that Aspergillus niger should be abbreviated, but only after it is mentioned in each section. Can you please check that you write Aspergillus niger once in each paragraph and A. niger in the rest.
Author Response
Dear Reviewer,
Thank you so much for appreciating our hard work. We will always be grateful for your valuable comments that have given our manuscript a good shape.
Stay safe.
